# *ALDH3A2*, *ODF2*, *QSOX2*, and MicroRNA-503-5p Expression to Forecast Recurrence in *TMPRSS2-ERG*-Positive Prostate Cancer

**DOI:** 10.3390/ijms231911695

**Published:** 2022-10-02

**Authors:** Anastasiya A. Kobelyatskaya, Alexander A. Kudryavtsev, Anna V. Kudryavtseva, Anastasiya V. Snezhkina, Maria S. Fedorova, Dmitry V. Kalinin, Vladislav S. Pavlov, Zulfiya G. Guvatova, Pavel A. Naberezhnev, Kirill M. Nyushko, Boris Y. Alekseev, George S. Krasnov, Elizaveta V. Bulavkina, Elena A. Pudova

**Affiliations:** 1Engelhardt Institute of Molecular Biology, Russian Academy of Sciences, 119991 Moscow, Russia; 2Vishnevsky Institute of Surgery, Ministry of Health of the Russian Federation, 117997 Moscow, Russia; 3National Medical Research Radiological Center, Ministry of Health of the Russian Federation, 125284 Moscow, Russia

**Keywords:** prostate cancer, primary tumor, recurrence, TMPRSS2-ERG, subtype, expression, RNA-Seq, miRNA-Seq, forecasting, neural network

## Abstract

Following radical surgery, patients may suffer a relapse. It is important to identify such patients so that therapy tactics can be modified appropriately. Existing stratification schemes do not display the probability of recurrence with enough precision since locally advanced prostate cancer (PCa) is classified as high-risk but is not ranked in greater detail. Between 40 and 50% of PCa cases belong to the TMPRSS2-ERG subtype that is a sufficiently homogeneous group for high-precision prognostic marker search to be possible. This study includes two independent cohorts and is based on high throughput sequencing and qPCR data. As a result, we have been able to suggest a perspective-trained model involving a deep neural network based on both qPCR data for mRNA and miRNA and clinicopathological criteria that can be used for recurrence risk forecasts in patients with TMPRSS2-ERG-positive, locally advanced PCa (the model uses *ALDH3A2* + *ODF2* + *QSOX2* + hsa-miR-503-5p + ISUP + pT, with an AUC = 0.944). In addition to the prognostic model’s use of identified differentially expressed genes and miRNAs, miRNA–target pairs were found that correlate with the prognosis and can be presented as an interactome network.

## 1. Introduction

Prostate cancer (PCa), including the localized (pT2) and locally advanced (pT3) forms, is one of the most common cancers in men. An important clinical task is setting optimal treatment strategies for locally advanced PCa. During this stage, the tumor grows into the prostate capsule, and, although distant metastases are absent, metastases to regional lymph nodes (lymphatic disseminations) are possible. The therapeutic concept in patients with PCa is determined by the prevalence of the oncological process. In patients with localized and locally advanced disease forms, the main methods of intervention are radical prostatectomy and radiation therapy. Following either of these, hormone therapy is the key method of treatment for patients with metastatic PCa or recurrent PCa. However, hormone therapy has only a temporary effect, lasting an average of 1.5–2 years. Then, despite the castration level of testosterone, the tumor process begins to progress, and the disease switches over to the stage of castration-resistant PCa, characterized by an unfavorable prognosis and aggressive course of the disease. After radical prostatectomy with pelvic lymph node dissection, some patients may suffer a relapse [1,2,3]. Consequently, it is important to identify this group of patients in order to modify their treatment strategy.

To date, there are several schemes for stratification of patients with PCa. According to the classification proposed by D’Amico et al. [4], there are three risk groups: low, intermediate, and high. Patients are stratified by clinicopathological parameters, such as their Gleason score, assessment of tumor extension (category T), the presence of lymphatic dissemination, and the level of serum preoperative prostate-specific antigen (PSA). Unfortunately, these three defined risk groups reflect the probability of recurrence with insufficient accuracy [5,6]. Moreover, all cases of locally advanced PCa are classified as high-risk but without being subdivided in greater detail. In addition to such clinicopathological patient stratification, various panels of expression markers have now been developed. For instance, OncotypeDX GPS^®^ is designed to identify the low-risk group, while Prolaris^®^ indicates localized PCa [7]. Beyond these, the Decipher^®^ panel is provided primarily to assess the likelihood of high-risk tumor metastasis. Notwithstanding the above, none of the panels described are specifically designed to estimate the risk of recurrence for patients with locally advanced PCa. Therefore, for any clinical situation, it is necessary to choose and employ the optimal panel of markers with the highest sensitivity and specificity of testing.

The search for high-precision prognostic markers is significantly hampered by tumor heterogeneity between patients. Division of patients into groups that are more homogeneous by their molecular genetic properties makes it possible to form a more suitable cohort for marker identification [8,9]. Based on The Cancer Genome Atlas (TCGA) project, seven molecular subtypes of PCa have now been described [10]. Four subtypes are known, characterized by fused transcripts (*TMPRSS2-ERG*, *TMPRSS2-ETV1*, *TMPRSS2-ETV4*, and *TMPRSS2-FLI1*, with frequencies of 46%, 8%, 4%, and 1%, respectively), and three by gene point mutations (*SPOP*, *FOXA1*, or *IDH1*; 11%, 3%, and 1%, respectively) [11].

Between 40 and 50% of PCa cases belong to the TMPRSS2-ERG molecular subtype, which is characterized by fusion of the *TMPRSS2* and *ERG* genes [11]. It is believed that the fusion of *TMPRSS2* and *ERG* is an early event and found in prostatic intraepithelial neoplasia [12]. The *TMPRSS2-ERG* fusion transcript, according to several authors, is a potential marker of unfavorable prognosis, i.e., the risk of PCa recurrence [13,14,15]. The first signal of its onset is a biochemical recurrence, when the PSA level after radical prostatectomy increases by more than 0.2 ng/mL.

Currently, microRNAs (miRNAs) are of considerable interest both for understanding the molecular basis of pathogenesis and as diagnostic markers for various diseases, including cancers. The most interesting aspect of the expansion of the role of miRNAs in the study of cancers is their regulation of gene expression by targeting mRNA moieties [16]. A recent review on the role of miRNAs in carcinogenesis [17] noted that miR-205 and miR-338-3p can inhibit apoptosis of PCa cells by targeting *BCL2* mRNA [18]. In addition to miRNAs, long non-coding RNAs (lncRNA) are also gaining more and more interest in issues related to the pathogenesis and course of oncological diseases. A recent study has shown that *LINCRNA-p21* inhibits expression of many genes in a p53-dependent transcriptional response by inhibiting PCa cell proliferation [19]. Furthermore, interactome investigation using transcriptome data is becoming increasingly important in the study of the molecular mechanisms of carcinogenesis [20], but there have only been a few studies of this type.

Thus, identification of prognostic markers for the TMPRSS2-ERG subtype of PCa with enough predictive power will allow optimization of treatment strategies for this group of patients in line with the concept of personalized medicine. The current study is focused on both miRNA and mRNA expression, as well as on their interactions associated with the risk of PCa recurrence.

## 2. Results

### 2.1. Differentially Expressed miRNAs

Based on our miRNA-Seq data observed for Cohort-A, 214 differentially expressed (DE) miRNAs were found between the biochemical recurrence (BCR) and biochemical recurrence-free (BRF) groups in *TMPRSS2-ERG*-positive PCa (*p*-value ≤ 0.05, QLF test, Figure 1, Appendix A). In the corresponding comparison groups for Cohort-B, 80 such DE miRNAs were found (Figure 1, Appendix A).

The seven miRNAs, which were DE in both cohorts, are presented in a Venn diagram (Figure 2, Appendix A). However, only two (hsa-miR-200b-3p and hsa-miR-503-5p) of these seven genes had a high level and stable distribution of expression among the compared groups for further study (Table 1 and Appendix A, Appendix A).

### 2.2. Interactome Network miRNA–Target Correlations Associated with BCR

Using these miRNA-Seq data, as well as RNA-Seq data obtained in our previous study [21], we performed Spearman correlation analysis for the miRNAs and their targets according to predicted and validated databases among the BCR and BRF groups. Indeed, 77 miRNA–target pairs (41 unique miRNAs and 75 unique targets) were differentially correlated and supplemented with PPI and TF–target interactions (Figure 3, Appendix A, with the dynamic networks presented in Appendix A).

Considering the members of the network, it must be mentioned that only a few miRNAs and genes within the network are DE, namely hsa-miR-1-3p and *REEP1* (DE for both cohorts), while hsa-miR-146b-3p and hsa-miR-375-3p, as well as *ASPM*, *MMP24*, *OLFML2A*, *RAB29,* and *SPEF2* are DE only in Cohort-A, and hsa-miR-134-5p, hsa-miR-146a-5p, hsa-miR-32-5p, hsa-miR-505-3p, and hsa-miR-96-5p, with genes *ABCC5*, *ACSL4*, *CDK6*, *HOOK2*, *HS3ST3B1*, *ID4*, *ITM2A*, *PALLD*, *PPM1L*, *SHC1*, *SLC7A5*, *SMAD3*, *TELO2*, *TMX4*, and *TRHDE* are DE only in Cohort-B.

Pathway enrichment analysis was performed for all participants of the interactome network, as a result of which 107 significantly enriched pathways were identified (Appendix A), of which 18 have miRNAs involved in addition to the genes (Table 2).

### 2.3. Predictive Model of BCR Based on mRNA and miRNA Expression

Considering the results of our previous work, we selected genes for the creation of a combined model with the newly identified miRNA DE and clinicopathological parameters. At the first stage, a prognostic model was created and tested, consisting only of clinicopathological parameters, such as the Gleason score, ISUP criterion, and the pT and preoperative PSA levels. The training dataset was Cohort-A, and the validation dataset was Cohort-B. As a result, this predictive model had a sensitivity of 0.67, specificity 0.61, and AUC 0.631 for the validation dataset, which is an unacceptably weak output (Table 3, Figure 4).

Further models based on the RNA-Seq expression data for four genes (*ALDH3A2*, *CHKA*, *ODF2*, and *QSOX2*) and miRNA-Seq expression data of two miRNAs (hsa-miR-503-5p and hsa-miR-200b-3p) were created and tested. The most prospective predictive model to forecast the biochemical recurrence when looking at the primary tumor of locally advanced PCa was that based on changes in the expression of four mRNAs and one miRNA (*ALDH3A2* + *CHKA* + *ODF2* + *QSOX2* + hsa-miR-503-5p, Figure 4, Table 3 and Appendix A).

However, when using an ISUP criterion extension of the model as an additional predictor, we observed an improvement in the power of the predictive model (*ALDH3A2* + *CHKA* + *ODF2* + *QSOX2* + hsa-miR-503-5p + ISUP, Figure 4, Table 3 and Appendix A).

### 2.4. mRNA and miRNA Expression Validation by qPCR

The relative levels of expression of the mRNAs (*ALDH3A2*, *CHKA*, *ODF2*, and *QSOX2*) and miRNA (hsa-miR-200b-3p and hsa-miR-503-5p) were evaluated by qPCR in extended Cohort-A samples belonging to the TMPRSS2-ERG subtype with known biochemical recurrence status (n = 31). Significant alterations in the relative expressions (*p*-value ≤ 0.05 according to the MW test and Spearman correlation, Figure 5) between the BCR and BRF groups for the *ALDH3A2*, *ODF2,* and *QSOX2* genes, as well as for the hsa-miR-200b-3p and hsa-miR-503-5p miRNAs, were confirmed (Table 4). Since the change in the relative expression of the *CHKA* gene between groups was not significant according to the qPCR data, it was excluded from further analysis.

Based on qPCR data for the *ALDH3A2*, *ODF2*, and *QSOX2* mRNAs, has-miR-200b-3p and hsa-miR-503-5p miRNAs, as well as on clinicopathological parameters, predictive models were created and tested. The training dataset was the group of *TMPRSS2-ERG*-positive cases with a known status of biochemical recurrence and previously sequenced (n = 14), with the remaining samples that were not sequenced (n = 17) serving as the test dataset.

The combination of predictors *ALDH3A2* + *ODF2* + *QSOX2* + hsa-miR-503-5p + ISUP + pT proved to be the most promising, with a sensitivity of 0.89, specificity 1, and AUC 0.944 for the test dataset (Appendix A). At the same time, it should be emphasized that all models omitting any one of the predictors were much inferior in predictive power to the above prognostic model (*ALDH3A2* + *ODF2* + *QSOX2* + hsa-miR-503-5p + ISUP + pT, Figure 6, Table 5).

Thus, based on the relative expression of mRNAs and miRNAs obtained by qPCR in the primary tumor, the most prospective markers were identified and a prognostic model (*ALDH3A2* + *ODF2* + *QSOX2* + hsa-miR-503-5p + ISUP + pT) was constructed, designed to forecast the recurrence risk for *TMPRSS2-ERG*-positive, locally advanced PCa.

## 3. Discussion

After radical prostatectomy with pelvic lymph node dissection, some patients with locally advanced PCa may undergo recurrence, for which the main curative approach is hormone therapy, but this can still lead to the stage of castration-resistant PCa within a couple of years [1,2,3].

Determination of patients with recurrence potential in the primary tumor is important to allow modification of the treatment strategy. Biomarkers based on mRNA and miRNA expression can facilitate such recognition. However, identifying prognostic markers is complicated by the heterogeneity among patients with PCa. Focusing on a more homogeneous group, collated by their molecular genetic properties (PCa subtype-TMPRSS2-ERG), has made it possible to identify markers with high predictive power. When looking for a solution to this problem, we previously analyzed gene expression [21]. In the current work, we focused not only on miRNA or mRNA expression but also on differences in the interactome using transcriptomic data.

Compared with the results of DE genes (388 DE genes), quite a few DE miRNAs were found here (seven DE miRNAs) that showed significant differences between the unfavorable and favorable prognosis groups in both cohorts. It is known that the study of molecular genetic alterations to distinguish a primary tumor with recurrent potential from tumors without it is an extremely difficult task. Despite this, we succeeded in identifying two miRNAs (hsa-miR-200b-3p and hsa-miR-503-5p), the alteration of expression of which was confirmed by qPCR, and it was found expedient to use them as predictors in conjunction with assessment of mRNA expression in the prognostic model.

It was notable that the two identified miRNAs have previously been described in PCa and in other cancers. A recent study demonstrated that a change in the level of expression of the miRNA hsa-miR-503-5p is associated with the degree of differentiation of tumor cells in PCa [22]. Upregulation of hsa-miR-503-5p in BCR cases may be associated with a decrease in the expression of *PTEN*, *TIMP3,* and *PDCD4* and activation of the AKT signaling pathway [22]. TIMP3 is an inhibitor of *MMP*, and, when it is suppressed, *MMP2* is activated, this being involved in the destruction of the extracellular matrix, which, therefore, probably promotes cell motility, inducing migration and invasion [22,23]. Moreover, we found upregulation of hsa-miR-200b-3p, while a recent study detected that hsa-miR-200b-3p knockdown inhibits PCa cell proliferation and promotes cell apoptosis [24], although the mechanisms of this effect are not yet known. Correspondingly, several other studies have reported an increased expression of hsa-miR-200b-3p in PCa compared with that in benign prostatic hyperplasias [25], as well as a positive correlation with *TIMP4*, which is an inhibitor of *MMP2* [26,27]. Therefore, according to the literature data, the two identified upregulated miRNAs (hsa-miR-503-5p and hsa-miR-200b-3p) for the unfavorable prognosis group in *TMPRSS2-ERG*-positive, locally advanced PCa may be involved in an increase in cell motility and an inhibition of apoptosis, promoting cell proliferation, invasion, and migration.

The current study showed that analysis of differential miRNA expression between PCa prognoses offered relatively few miRNAs applicable as markers. In addition, we have suggested using an interactome approach in order to search for transcriptomic disturbance associated with unfavorable prognoses. Herewith, the analysis was mainly focused on the correlation of miRNA–target pairs and the differences in these correlations depending on the prognosis group. We have presented the most significant interactome differences between the prognosis groups in the form of an interaction network. It is noteworthy that the majority of miRNAs and genes of the compiled network are not DE between the prognosis groups. Based on this, it can be assumed that most of the displayed different miRNA–target correlations are associated with unfavorable prognosis, not due to the differential expression of an individual member of the pair but due to their interaction that occurs in the unfavorable prognosis group, although, at this stage of the work, the reasons for this association emerging remain unknown, but it is relevant since it is a further element of the puzzle of PCa pathogenesis. Enrichment analysis of the constructed network showed that the participants in the above differentially correlated pairs are those mainly involved in regulation of the cytoskeleton, in organization of the extracellular matrix, modulation of the immune response, and in differentiation, proliferation, apoptosis, and cell motility.

In our previous work, we offered a model based solely on gene expression (RNA-Seq data only) created using the random forest method. In the current work, we have suggested, on the one hand, a more complex, and, on the other hand, an improved model, which is also based on qPCR data and deep neural networks. As we have demonstrated in the results, clinicopathological model quality metrics show that, on their own, these characteristics of PCa do not have sufficient predictive power and cannot serve as independent markers for assessing the risk of recurrence when studying the primary tumor. However, it is rational to use them in combined models as additional predictors. The combination of expression sequencing data of mRNA and miRNAs allowed us to obtain a model that copes quite well with the task (AUC = 0.963), and, when clinicopathological criteria were included, it was possible to improve its quality ratings (AUC = 0.982), while qPCR confirmed the model’s predictive power (AUC = 0.944). As a result, the *ALDH3A2* + *ODF2* + *QSOX2* + hsa-miR-503-5p + ISUP + pT model, which combines both expression data of three mRNAs, the expression of one miRNA, and the data from two clinicopathological criteria, turned out to be the most promising for predicting the probability of recurrence from study of the primary tumor.

It should be noted that the genes for the model were selected very specifically, being among the best for this purpose according to the results of statistical tests and their involvement in key cellular processes. For example, *ALDH3A2* catalyzes the oxidation of long-chain aliphatic aldehydes to fatty acids. For the first time, the association of *ALDH3A2* gene expression with PCa has been established; namely, a decrease in *ALDH3A2* expression is associated with an unfavorable prognosis, which may be linked to a shift to the M2 phenotype, and, as a result, suppression of the immune function of the tumor microenvironment. In several studies, *ALDH3A2* downregulation has been associated with metastasis of colorectal cancer [28], as well as with adenocarcinoma of the esophagus [29]. Although the role of *ODF2* in carcinogenesis remains unclear at present, it is known that ODF2 is involved in the formation of the bipolar spindle [30] and limits the accumulation of β-catenin in the centrosome, which, in turn, prevents aberrant centrosome cleavage and spindle formation. It is possible that *ODF2* upregulation leads to impaired spindle formation and to chromosomal instability in the recurrence of *TMPRSS2-ERG*-positive, locally advanced PCa. It is known that *QSOX2* expression is regulated by transcription factor E2F1 [31] during the cell cycle and functions in the G1 phase [32]. *QSOX2* upregulation in an unfavorable prognosis can probably be associated with the passage of cells in the S- and G2-phases of the cell cycle instead of apoptosis, and, as a consequence, with tumor development. Several other investigations have shown an increase in *QSOX2* expression in colorectal cancer compared to normal tissue [33,34], which has been associated with poor prognoses [33] and poor overall survival [34]. Herewith, the issue of studying other molecular subtypes remains no less relevant. Although their frequency of occurrence among patients with PCa is lower (compared to the TMPRSS2-ERG subtype), that makes it difficult to form a sufficient dataset for the study. Another important point is that, in addition to the described subtypes, there are other genomic disturbances in parallel. For example, the loss of *PTEN*, *RB1*, and *TP53* genes is mostly specific to *ETS*-fusion cases, while the loss of *CHD1* and the increase in *SPINK1* are observed in *ETS*-negative subtypes. In addition, mutations in the *ATM* and *BRACA2* genes have been described, as well as in the DNA damage repair (DDR) and double strand break (DSB) repair genes. Changes in these genes are found in primary PCa, but only in 20%, and the enrichment of these mutations is found already at the stage of metastatic and castration-resistant PCa. The described genomic events are of interest as a base for further comprehensive study of the mechanisms of PCa progression.

As a result of our work, a promising trained model has been suggested in the form of a deep neural network based on qPCR data for mRNA and miRNA and clinicopathological characteristics (*ALDH3A2* + *ODF2* + *QSOX2* + hsa-miR-503-5p + ISUP + pT, AUC = 0.944) to forecast the risk of recurrence in patients with *TMPRSS2-ERG*-positive locally advanced PCa. In addition to the prognostic model, the identified DE genes, and miRNAs, several pairs of miRNAs–targets were found, the correlation of which is also associated with the prognosis and presented as an interactome network.

## 4. Materials and Methods

### 4.1. Materials

The study used two cohorts: (1) extended collection of PCa tumor samples obtained from Russian patients (RNA-Seq was previously performed for part of the samples) [21]—Cohort-A, and (2) high-throughput sequencing data of the TCGA-PRAD project—Cohort-B. The PCa tumor samples were characterized at the National Medical Research Radiological Center (Ministry of Health of the Russian Federation). Each sample contained a minimum of 70% tumor cells. Samples of both cohorts were from patients not undergoing neoadjuvant therapy and belonging to the Caucasian population, for which the disease recurrence status and TMPRSS2-ERG molecular subtype condition were known (Table 6) [10,21].

### 4.2. Methods

#### 4.2.1. Isolation of RNA and Reverse Transcription

Total RNA was isolated from fresh frozen tumor tissue samples using the MagNA Pure Compact RNA Kit (Roche, Basel, Switzerland) on a MagNA Pure Compact System (Roche) according to the manufacturer’s protocol. The concentration of isolated RNA was determined on a Qubit 2.0 fluorimeter (Thermo Fisher Scientific, Waltham, MA, USA) using the Qubit RNA BR Assay Kit (Thermo Fisher Scientific). The quality of the isolated RNA was assessed on an Agilent 2100 bioanalyzer (Agilent Technologies, Santa Clara, CA, USA) using an Agilent RNA 6000 Nano Kit (Agilent Technologies). Reverse transcription of total RNA was performed using Mint reverse transcriptase (Evrogen, Moscow, Russia). The resulting cDNA samples were diluted 100-fold with 0.1X TE buffer. The reverse transcription reaction on the miRNA template was carried out using the TaqMan Advanced miRNA cDNA Synthesis Kit (Thermo Fisher Scientific). The resulting cDNA samples were diluted 10-fold with 0.1X TE buffer.

#### 4.2.2. miRNA Sequencing

MiRNA libraries were prepared for 24 PCa samples (those for which RNA-Seq was previously performed, Cohort-A) using the NEBNext Small RNA Library Prep Set for Illumina (New England Biolabs, Ipswich, MA, USA) for Illumina according to the manufacturer’s protocol. Sequencing was performed on a NextSeq500 system (Illumina, San Diego, CA, USA) using a NextSeq 500/550 High Output Kit v2.5 (Illumina), read length 36 nt, read mode single, coverage about 10 million per sample.

#### 4.2.3. Quantitative PCR (qPCR)

The TaqMan Gene Expression Assays Hs03063375_ft (Thermo Fisher Scientific) kit was used to assess the expression level of the *TMPRSS2-ERG* fusion transcript. The level of gene expression was assessed using primers, the sequences of which are listed in Table 7. The miRNA expression level was assessed using TaqMan™ Advanced miRNA Assay kits (Thermo Fisher Scientific): 477999_mir (hsa-miR-28-3p), 477963_mir (hsa-miR-200b-3p), 478143_mir (hsa-miR-503-5p). The *RPN1* gene [35] was used as a control gene for analysis of the relative expression (primer and probe sequences for the *RPN1* gene [36]). Further, hsa-miR-28-3p miRNA was used as a control miRNA for analysis of the relative expression. Real-time PCR was performed using the Applied Biosystems 7500 Real-Time PCR System (Thermo Fisher Scientific). Each PCR reaction was performed in three technical repetitions. The following program was used for mRNA amplification: 95 °C for 15 min, 40 cycles 95 °C for 15 s, 60 °C for 60 s. To assess the level of expression, the method of relative measurements (*Δ*CT) was used, and calculations were performed using the ATG program (Analysis of Transcription of Genes).

#### 4.2.4. Data Analysis

MiRNA-Seq data were processed by the miRge 3.0 pipeline (Baltimore, MD, USA) [37]. Differential expression analysis was performed in the R environment (v.3.6.3) [38] using the edgeR package (v.3.24.3) [39]. To normalize the obtained data, the TMM (Trimmed Mean of M-values) method was applied with calculation of the CPM (counts per million) considering the normalization coefficients. The quasi-likelihood F-test (QLF) and Mann–Whitney U-test (MW) were used to evaluate the significance of changes in miRNA expression. Benjamini–Hochberg correction was applied to calculate the FDR for all tests. MiRNA passing *p*-value QLF or MW ≤ 0.05 were considered differentially expressed and annotated using the R package multiMiR (v.1.8.0, Novato, CA, USA) [40] and Encori database [41].

Validated (miRecords, miRTarBase, TarBase) and predicted (Diana-microT, ElMMo3, MicroCosm, miRanda, miRDB, PicTar, PITA, TargetScan, RNA22, miRmap) databases were used to identify the miRNA targets. For miRNA–target pairs, a Spearman (*r_s_*) and a Pearson (*r_p_*) correlation analysis were performed using the cor.test R function, and the coefficient of determination (adjusted R^2^) was calculated using the linear regression method (lm R function). Differential correlation was calculated as the difference between the correlation coefficients of the two groups (BCR group *r_s_* minus BRF group *r_s_*). Fisher’s *z* (Fisher’s *r*-to-*z* approach) was used to determine the significance of the difference.

Transcriptional factor (TF) targets were identified using the CHIP atlas database [42]. To establish protein–protein interactions (PPI), the STRING network database (physical_v11.5) [43] was used. According to the databases of PPI and TF targets, to display the completeness of the picture of interactions, a similar analysis of the Spearman and Pearson correlation analyses and linear regression was carried out. For selected miRNAs and their targets, an overrepresentation analysis (ORA) of biological pathways was performed for the GO [44], KEGG [45], and Reactome [46] pathway databases using the clusterProfiler [47] and ReactomePA [48] packages, as well as the miRwalk database [49]. Pathway enrichment was considered significant at a *p*-value < 0.05. The interactome network (miRNA–target, lncRNA–target, PPI, TF–target) was visualized using the visNetwork package [50].

Predictive models were created using the machine learning method-neural networks. To build a fully connected neural network (FCNN), the keras [51], tensorflow [52], and kerasR [53] libraries were used. Before training the models, the input data were scaled and centered using the scale R function. FCNN had the following architecture: an input layer (n predictors), seven hidden layers (with a relu/tanh activation function and 20–150 neurons), and an output layer consisting of two neurons and having a softmax activation function. The gradient calculation method was backpropagation, the loss function was binary crossentropy, and the optimizer was the Adam algorithm [54], with the learning rate parameter = 0.003. The model quality metrics were: sensitivity, specificity, normalized proportion of correct answers (Cohen’s kappa), and precision. The reference value for these model quality metrics was one. The constructed models were subject to ROC analysis (receiver operating characteristic, R package pROC) [55], which allows assessment of the quality of the classification produced by the model by calculating the area under the error curve (AUC).

## Figures and Tables

**Figure 1 ijms-23-11695-f001:**
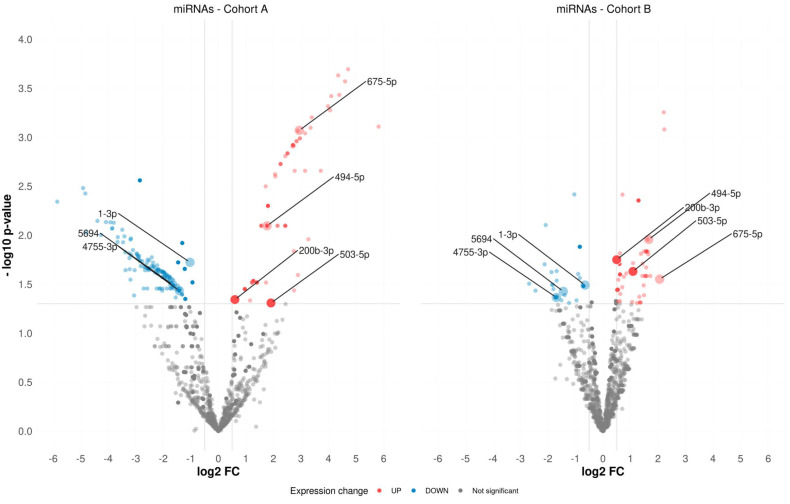
Volcano plot of differentially expressed miRNAs in both cohorts.

**Figure 2 ijms-23-11695-f002:**
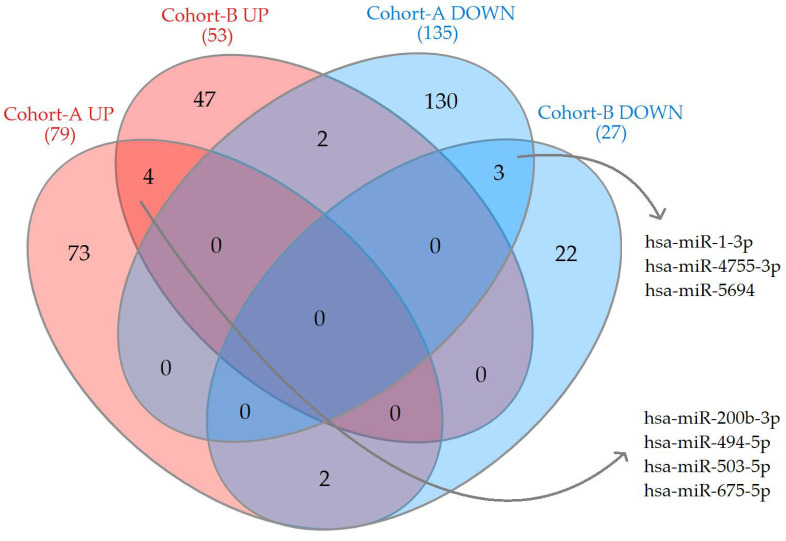
Venn diagram of upregulated and downregulated miRNAs in both cohorts.

**Figure 3 ijms-23-11695-f003:**
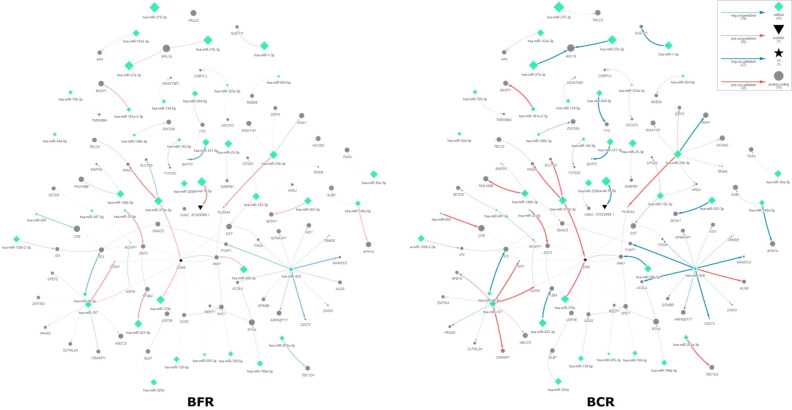
Interactome networks for BCR/BRF groups of the PCa TMPRSS2-ERG molecular subtype. BCR—biochemical recurrence group, BRF—biochemical recurrence-free group.

**Figure 4 ijms-23-11695-f004:**
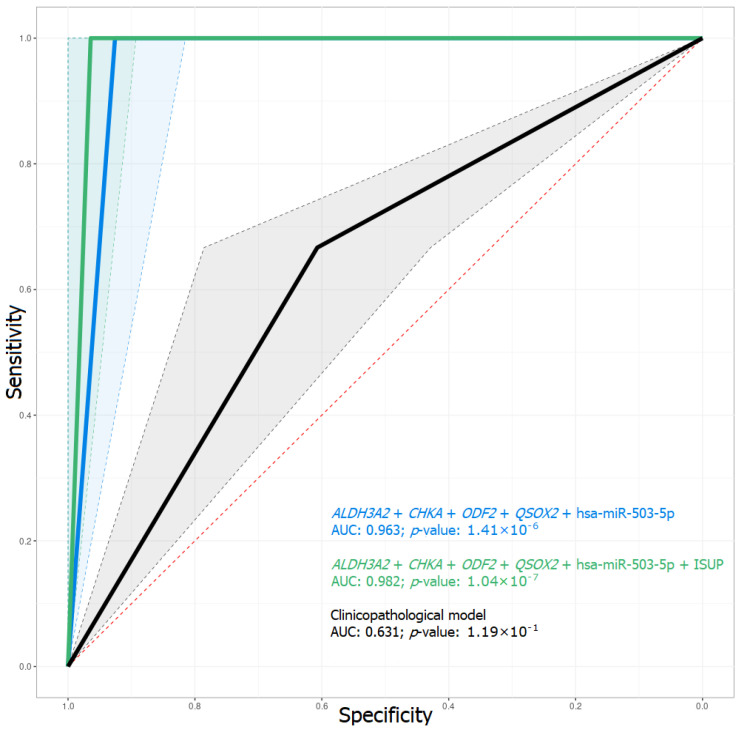
ROC-curve of predictive models of the test dataset. AUC—area under the error curve.

**Figure 5 ijms-23-11695-f005:**
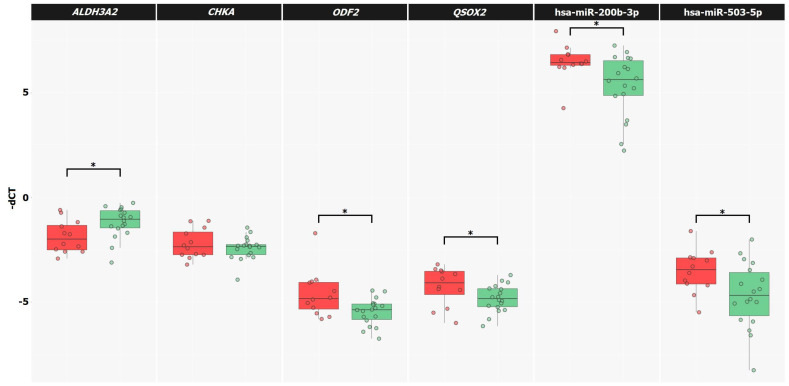
The relative expression of mRNA and miRNA for the unfavorable prognosis group is marked in red, and, for the favorable prognosis group, green. *—significant alteration.

**Figure 6 ijms-23-11695-f006:**
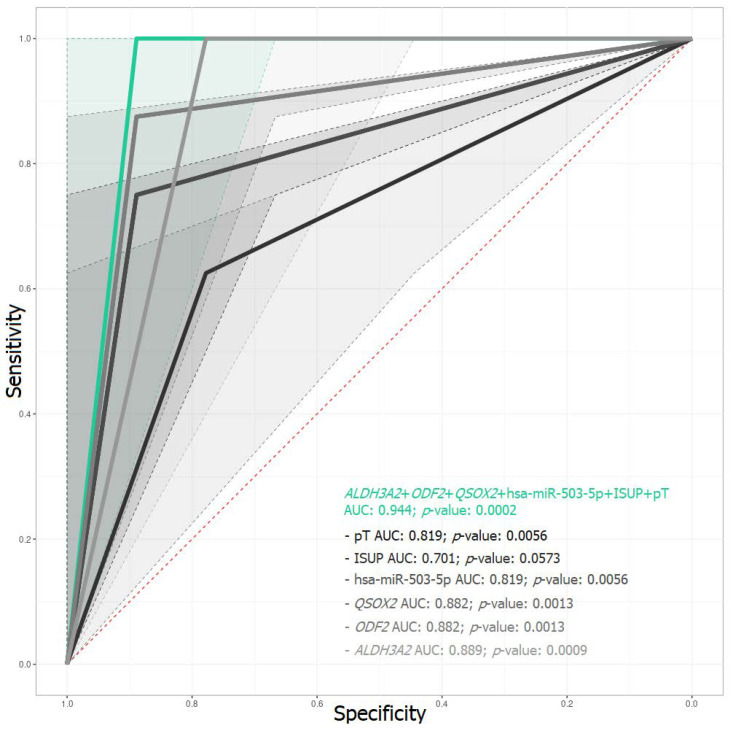
ROC-curve of predictive models of test dataset. The *ALDH3A2* + *ODF2* + *QSOX2* + hsa-miR-503-5p + ISUP + pT model is colored in green; the models with a predictor omitted are shown in gray shades. AUC—area under the error curve.

**Table 1 ijms-23-11695-t001:** Upregulated miRNAs in unfavorable prognosis group.

miRNAs	FC	log2CPM	QLF *p*-Value	MW *p*-Value
Cohort-A				
hsa-miR-503-5p	↑3.75	2.36	4.9 × 10^−2^	8.5 × 10^−4^
hsa-miR-200b-3p	↑1.52	11.89	4.5 × 10^−2^	4.9 × 10^−2^
Cohort-B				
hsa-miR-503-5p	↑2.12	2.64	2.3 × 10^−2^	2.6 × 10^−2^
hsa-miR-200b-3p	↑1.41	9.61	1.8 × 10^−2^	2.7 × 10^−2^

**Table 2 ijms-23-11695-t002:** Enriched pathways for network pairs.

Pathway Name (ID, Ontology): Genes, miRNAs	*p*-Value
KEGG	
TGF-beta signaling pathway (hsa04350): *ID3 ID4 SMAD3* miR-27a-3p	5.5 × 10^−3^
ErbB signaling pathway (hsa04012): *PAK1 SHC1 SOS2* let-7b-5p miR-26b-5p miR-429	4.1 × 10^−3^
Natural killer cell mediated cytotoxicity (hsa04650): *PAK1 SHC1 SOS2* let-7b-5p miR-26b-5p miR-429	1.2 × 10^−2^
Chemokine signaling pathway (hsa04062): *PAK1 SHC1 SOS2* let-7b-5p miR-26b-5p miR-429	3.7 × 10^−2^
Focal adhesion (hsa04510): *PAK1 SHC1 SOS2* let-7b-5p miR-26b-5p miR-429	4.1 × 10^−2^
Gene Ontology	
Negative regulation of neuron differentiation (GO:0045665, BP): *ASPM DNM3 ID3 ID4 RAB29 RTN4* miR-101-3p miR-148b-3p miR-26b-5p miR-27a-3p miR-34a-5p	2.0 × 10^−4^
Negative regulation of osteoblast differentiation (GO:0045668, BP): *CDK6 ID3 SMAD3* let-7b-5p miR-103a-3p miR-107 miR-26b-5p miR-30a-3p miR-34a-5p	7.4 × 10^−4^
Extracellular matrix organization (GO:0030198, BP): *FOXF1 GPM6B MMP24 NID1 OLFML2A SMAD3* miR-146a-5p	2.5 × 10^−3^
Ras protein signal transduction (GO:0007265, BP): *ARHGEF17 RAB29 SHC1 SOS2 ZNF304* miR-429	2.7 × 10^−2^
Regulation of axonogenesis (GO:0050770, BP): *PAK1 PLXNA4 RTN4* miR-101-3p miR-148b-3p miR-34a-5p	3.2 × 10^−2^
Negative regulation of cell growth (GO:0030308, BP): *PAK1 RTN4 SMAD3* miR-101-3p miR-148b-3p miR-34a-5p	3.3 × 10^−2^
Cellular response to insulin stimulus (GO:0032869, BP): *PAK1 SHC1 TBC1D4* let-7b-5p miR-26b-5p	4.9 × 10^−2^
Z disc (GO:0030018, CC): *DST PAK1 PALLD* let-7b-5p miR-26b-5p miR-32-5p	1.4 × 10^−2^
Ruffle (GO:0001726, CC): *CDK6 PAK1 PALLD* let-7b-5p miR-103a-3p miR-107 miR-26b-5p miR-30a-3p miR-34a-5p	3.0 × 10^−2^
Nuclear envelope (GO:0005635, CC): *DST OSBPL3 PAK1 RTN4 SMAD3* miR-101-3p miR-148b-3p miR-32-5p miR-34a-5p	3.3 × 10^−2^
Collagen binding (GO:0005518, MF): *NID1 PAK1 SMAD3* let-7b-5p miR-26b-5p	2.1 × 10^−3^
Transcription corepressor activity (GO:0003714, MF): *BASP1 BATF3 ID3 ID4* miR-26b-5p miR-27a-3p	1.3 × 10^−2^
RNA polymerase II transcription factor binding (GO:0001085, MF): *ID3 ID4 SMAD3* miR-27a-3p	2.1 × 10^−2^

**Table 3 ijms-23-11695-t003:** Metrics of predictive models. Se—sensitivity, Sp—specificity, Ka—kappa (normalized proportion of correct answers), Pr—precision, AUC—area under the error curve.

Models	Test Dataset/Training Dataset				
	Se	Sp	Ka	Pr	AUC
Based on clinicopathological parameters	0.67/0.75	0.61/1.00	0.60/0.93	0.27/1.00	0.631/0.875
*ALDH3A2* + *CHKA* + *ODF2* + *QSOX2* + hsa-miR-503-5p	1.00/1.00	0.93/1.00	0.94/1.00	0.75/1.00	0.963/1.000
*ALDH3A2* + *CHKA* + *ODF2* + *QSOX2* + hsa-miR-503-5p + ISUP	1.00/1.00	0.96/1.00	0.97/1.00	0.86/1.00	0.982/1.000

**Table 4 ijms-23-11695-t004:** Alteration in relative expression between BCR and BRF groups. FC—fold change in expression level, log2CPM—expression level in the cohort, *r_s_*—Spearman’s correlation coefficient, MW—Mann–Whitney U test. ↑—upregulation, ↓—downregulation. *—*p*-value ≤ 0.05.

mRNA/miRNA	FC	MW *p*-Value	Spearman Correlation	
			*r_s_*	*p*-Value
*ALDH3A2*	↓1.56	0.023 *	−0.42	0.019 *
*CHKA*	↑1.22	0.611	0.10	0.605
*ODF2*	↑2.45	0.019 *	0.43	0.017 *
*QSOX2*	↑1.58	0.049 *	0.35	0.045 *
hsa-miR-200b-3p	↑1.73	0.025 *	0.42	0.022 *
hsa-miR-503-5p	↑1.72	0.035 *	0.39	0.032 *

**Table 5 ijms-23-11695-t005:** Metrics of predictive models. Se—sensitivity, Sp—specificity, Ka—kappa (normalized proportion of correct answers), Pr—precision, AUC—area under the error curve.

Models	Test Dataset/Training Dataset				
	Se	Sp	Ka	Pr	AUC
*ALDH3A2* + *ODF2* + *QSOX2* + hsa-miR-503-5p + ISUP + pT	0.89/1.00	1.00/1.00	0.88/1.00	1.00/1.00	0.944/1.000
*ALDH3A2* + *ODF2* + *QSOX2* + hsa-miR-503-5p + ISUP	0.89/1.00	0.75/1.00	0.64/1.00	0.80/1.00	0.819/1.000
*ALDH3A2* + *ODF2* + *QSOX2* + hsa-miR-503-5p + pT	0.78/1.00	0.62/1.00	0.41/1.00	0.70/1.00	0.701/1.000
*ALDH3A2* + *ODF2* + *QSOX2* + ISUP + pT	0.89/1.00	0.75/1.00	0.64/1.00	0.80/1.00	0.819/1.000
*ALDH3A2* + *ODF2* + hsa-miR-503-5p + ISUP + pT	0.89/1.00	0.88/1.00	0.76/1.00	0.89/1.00	0.882/1.000
*ALDH3A2* + *QSOX2* + hsa-miR-503-5p + ISUP + pT	0.89/0.89	0.88/1.00	0.76/0.86	0.89/1.00	0.882/0.944
*ODF2* + *QSOX2* + hsa-miR-503-5p + ISUP + pT	0.78/1.00	1.00/1.00	0.77/1.00	1.00/1.00	0.889/1.000

**Table 6 ijms-23-11695-t006:** Clinical and pathological characteristics of the studied cohorts. pT—primary tumor estimation, N—regional lymph nodes, M—distant metastases, R—residual tumor estimation.

Criterion		Cohort-A		Cohort-B	
		n	%	n	%
PCa samples		111	100	154	100
Age, years		63 (41–73)	-	62 (46–78)	-
pT	pT3a	55	50	85	55
	pT3b	52	47	65	42
	pT4	4	3	4	3
pN	pN0	73	66	102	66
	pN1	38	34	42	27
cM	cM0	111	100	154	100
	cM1	0	0	0	0
Gleason score	6	15	14	6	4
	7	62	56	59	36
	8	13	12	23	16
	9	20	18	65	43
	10	1	0	1	<1
ISUP	1	15	14	6	4
	2	30	27	28	18
	3	32	28	31	20
	4	13	12	23	15
	5	21	19	66	43
PSA, ng/ml		13.6 (2.5–61)	-	8.7 (2.7–87)	-
Biochemical recurrence (PSA ≥ 0.2 ng/mL)	Yes, N0R0	22	20	15	10
	Yes, N1/R1	6	5	29	19
	No, any N/R	57	51	104	67
	unknown	26	24	6	4
TMPRSS2-ERG molecular subtype	Yes	52	47	76	49
	No	59	53	78	51

**Table 7 ijms-23-11695-t007:** Primer sequences for assessing the level of mRNA expression.

mRNA	Primer Sequence (5′ → 3′)	Product Length, b.p.
*ALDH3A2*	F: GTCTGGACAAGAAGCGAGTAAG R: GACCATGAACGACTGATGAGAG	110
*CHKA*	F: GATCCGAACAAGCTCAGAAAGA R: CTGCGAGAATGGCAAACATAAC	84
*ODF2*	F: GGCCTGATTTGTATCTCTGGAA R: GATGAGGACTGGTTGGGTAAAG	124
*QSOX2*	F: CTTAGACCTGATCCCGTATGAAAG R: GTAACACGAAGGGACTGAAGAA	103

## Data Availability

All data generated or analyzed during this study are available within the article or upon request from the corresponding author.

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
