# Peer review of "ALDH3A2, ODF2, QSOX2, and MicroRNA-503-5p Expression to Forecast Recurrence in TMPRSS2-ERG-Positive Prostate Cancer"

_ijms, 2022, doi:10.3390/ijms231911695_

Round 1

Reviewer 1 Report

Kobelyatskaya et al., describe a multiple-biomarkers model that can be used for recurrence risk predicts in patients with TMPRSS2-ERG-positive PCa. The authors used two cohorts to discover the model, and validated RNA and miRNA expression using qPCR in one cohort. It addresses an important question, but its impact is limited without independent cohort validation.

Main points:

1.     The sample size is small, and the statistical tests did not perform multiple testing.

2.     Many miRNA shows very different expression levels between two cohorts such as hsa-miR-494-5p and hsa-miR-5694 highly variable in cohort B but hardly expressed in cohort A. Is this because sampling bias?

3.     Does common PC genomic subtypes, such as TP53, DNA repair defects, RB1 loss, PTEN loss, impact on the model please?

Author Response

Dear reviewer, thank you for carefully reading the article and for your comments.

Point 1: The sample size is small, and the statistical tests did not perform multiple testing.

Response 1: At the initial stage of research, the Cohort-A consisted of 111 cases of locally advanced PCa, and Cohort-B – 154, that were representative. At the same time, the TMPRSS2-ERG subtype occurs in half of PCa cases, and recurrence occurs in a third. One very important clinical point is that part of patients suffering a recurrence have a residual tumor (mainly in Cohort-B). Such cases had to be excluded from the group of unfavorable prognosis, since they would introduce an disturbance of the result. Since the results of statistical tests were based on p-value ≤ 0.05, we tested the expression by qPCR, that confirmed the differential expression for most of the identified potential predictors in extended Cohort-A.

Point 2: Many miRNA shows very different expression levels between two cohorts such as hsa-miR-494-5p and hsa-miR-5694 highly variable in cohort B but hardly expressed in cohort A. Is this because sampling bias?

Response 2: Some miRNAs had a change in expression in one cohort and not in another. It may be due to the presence of population differences, although other reasons are possible. Our study focused on finding differences that were not biased due to a particular cohort. We were looking for persistent changes among prognostic groups that were typical for both cohorts. As for miR-494-5p and miR-5694, it was not a sampling bias, since the results of both cohorts were concordant for these miRNAs (Table S1). The expression level of these two miRNAs in both cohorts is not enough (logCPM) at p-value ≤ 0.05 and a quite high FC (fold change). Such changes in expression level are extremely difficult to validate and interpret. It is especially difficult to perform it in clinical practice, since the capabilities of medical institutions are limited, and these markers will not be used. Perhaps the heatmap is not sufficient to display the results, and the visualizations has been updated and supplemented.

Point 3: Does common PC genomic subtypes, such as TP53, DNA repair defects, RB1 loss, PTEN loss, impact on the model please?

Response 3: The TMPRSS2-ERG gene fusion is considered an early event. While the aforementioned genomic changes are commonly typical for more advanced stages of PCa, namely for the metastatic form. Considering that the studied PCa was locally advanced form, during the work no such changes were observed at the expression level. Notwithstanding, the task of testing the association of the prognostic model with similar genomic events was not set within the current work. However, there are possible that changes preceding these events already exist, but this is only an assumption.

Reviewer 2 Report

In this manuscript, Kobelyatskaya et al. seek to unravel common genetic features of TMPRSS2-ERG-positive prostate cancer that can predict biochemical recurrence. The authors established a model involving ALDH3A2, ODF2, QSOX2, hsa-miR-503-5p, ISUP and pT as biomarkers, which can predict the replase of those patients. The manuscript is in general well written, and the newly established model can be of interest to prostate cancer clinicians and researchers. I only two minor comments as outlined below:

1. The current study is mainly focused on TMPRSS2-ERG-positive prostate cancer, but I don’t see any mentioning of TMPRSS2-ERG-negative prostate cancer at all throughout the text. Have the authors tried this model on TMPRSS2-ERG-negative prostate cancers, and if yes, what is the outcome?

2. The authors need to compare the power of their model against previous models mentioned in the introduction (i.e., OncotypeDX GPS®, Prolaris®, Decipher®) on their samples to prove that their predictive model is indeed better than the other ones.

Author Response

Dear reviewer, thank you for carefully reading the article and for your comments.

Point 1: The current study is mainly focused on TMPRSS2-ERG-positive prostate cancer, but I don’t see any mentioning of TMPRSS2-ERG-negative prostate cancer at all throughout the text. Have the authors tried this model on TMPRSS2-ERG-negative prostate cancers, and if yes, what is the outcome?

Response 1: It is known that PCa is quite heterogeneous among patients, and a few difficulties arise in its study. In our earlier studies, we have already described key changes between the TMPRSS2-ERG-positive and TMPRSS2-ERG-negative PCa cases. It should be mentioned that TMPRSS2-ERG-negative group is all other molecular subtypes of PCa, as well as cases that do not belong to any of the 7 described subtypes. Therefore, the TMPRSS2-ERG-negative group is too diverse to search for markers, so our task was to search for markers for the TMPRSS2-ERG subtype as a more homogeneous group. The suggested model was not tested on the TMPRSS2-ERG-negative group; these predictors are not intended for this group and are not differentially expressed.

Point 2: The authors need to compare the power of their model against previous models mentioned in the introduction (i.e., OncotypeDX GPS®, Prolaris®, Decipher®) on their samples to prove that their predictive model is indeed better than the other ones.

Response 2: It is extremely important to note that in our results we do not position the model as better than OncotypeDX GPS®, Prolaris®, Decipher®. Comparison of our predictive model proposed in this work with existing expression panels is not possible, since they are aimed at solving different clinical tasks. The OncotypeDX is designed for the low-risk group, Prolaris – for localized PCa, and Decipher – for tumor metastasis. But none of them is able to assess the risk of recurrence of locally advanced PCa. At the same time, the lack of an effective predictive model for solving the problem of predicting recurrence in the study of primary tumors of locally advanced PCa makes our results relevant for solving this issue.

Round 2

Reviewer 1 Report

More data shows DDR in lethal prostate cancer acquired at early stage, and DDR can lead to TMPRSS2-ERG fusion. DDR in prostate cancer is an important prognostic marker, should be considered as a confounding factor. If such experiment is not achievable, this should be discussed in the discussion as a limitation of the paper. 

Author Response

Dear reviewer,

We were able to find only one study aimed at describing the relationship between TMPRSS2-ERG fusion and DNA damage repair (35652618). At the same time, the study was performed on later stage of PCa, namely metastatic castration-resistant PCa, for which they are studying the response in patients to the Ra-223. In their work, they noted that TMPRSS2-ERG and RB mutations were associated with poor overall survival in this form of cancer.

In addition, a recent review considering at PCa subtypes (33850790) under the concept of future directions mentioned a possible association between TMPRSS2-ERG and other genomic events in metastatic PCa.

All of this is consistent with our suggestion that it is possible that similar disturbances might appear in later stages.

In conclusion, we would like to note that we find your comment about possible association of TMPRSS2-ERG fusion with other genomic events (PTEN, RB, DDR) incredibly valuable and promising for future research into the pathogenesis and progression of PCa.